# Application of Model-Based Time Series Prediction of Infrared Long-Wave Radiation Data for Exploring the Precursory Patterns Associated with the 2021 Madoi Earthquake

**Jingye Zhang [1], Ke Sun [1,*], Junqing Zhu [1], Ning Mao [1] and Dimitar Ouzounov [2]**

1   Institute of Earthquake Forecasting, CEA, Beijing 100036, China; zhangjy@ief.ac.cn (J.Z.);
    zhujq@ief.ac.cn (J.Z.); maoning@ief.ac.cn (N.M.)
2   Institute for Earth, Computing, Human and Observing, Chapman University, Orange, CA 92866, USA;
    ouzounov@chapman.edu
*   Correspondence: sunke@cea-ies.ac.cn

**Abstract:** Taking the Madoi $M_S$ 7.4 earthquake of 21 May 2021 as an example, this paper proposes using time series prediction models to predict the outgoing long-wave radiation (OLR) anomalies and study short-term pre-earthquake signals. Five time series prediction models, including autoregressive integrated moving average (ARIMA) and long short-term memory (LSTM), were trained with the OLR time series data of the aseismic moments in the $5° \times 5°$ spatial range around the epicenter. The model with the highest prediction accuracy was selected to retrospectively predict the OLR values during the aseismic period and before the earthquake in the area. It was found, by comparing the predicted time series values with the actual time series value, that the similarity indexes of the two time series before the earthquake were lower than the index of the aseismic period, indicating that the predicted time series before the earthquake significantly differed from the actual time series. Meanwhile, the temporal and spatial distribution characteristics of the anomalies in the 90 days before the earthquake were analyzed with a 95% confidence interval as the criterion of the anomalies, and the following was found: out of 25 grids, 18 grids showed anomalies—the anomalies of the different grids appeared on similar dates, and the anomalies of high values appeared centrally at the time of the earthquake, which supports the hypothesis that pre-earthquake signals may be associated with the earthquake.

**Keywords:** LSTM; OLR time series; Madoi earthquake; pre-earthquake signals

## 1. Introduction

An earthquake is a process of the rapid release of the accumulated Earth's tectonic stress, which breaks through the critical value of the elastic rupture of rocks. In the past few decades, with the help of a large amount of pre-earthquake remote sensing data, researchers have found that large earthquakes are often accompanied by various anomalies before they occur, such as anomalies in the atmospheric temperature and humidity, the content of gases, including $CH_4$ and CO, at active ruptures, the surface latent heat flux (SLHF), total electron content (TEC), outgoing long-wave radiation (OLR), etc., which change significantly before an earthquake occurs [1–5]. Among them, the OLR data can characterize the radiative changes in the whole geophysical system, which contains the radiative information of the cloud top, the surface, and the atmosphere above it; this characterization helps to capture the anomalous changes in the geopathic system in the pre-earthquake area, and it is an essential parameter for the study of pre-earthquake signals [6,7].

The study of OLR anomalies linked to earthquakes first started in China: Sun et al. [8] from the Qinghai Seismological Bureau explored the relationship between long-wave radiation and earthquakes in 1990. Since then, researchers worldwide have begun to conduct extensive pre-seismic anomaly studies using OLR data. They have summarized

several methods for OLR seismic information extraction, such as the vorticity background field method [9,10], the standard deviation thresholding method [11–13], and the method of the relative variance rate of power spectrum estimation [14,15]. The study of these methods proves that OLR has high potential for application in earthquake prediction.

The accumulation of OLR data over the years has significantly challenged traditional data processing algorithms. Rapid advancements in machine learning have shown promising results across various industries in recent years. The field of seismology has also benefited from the thriving development of machine learning. Experts and scholars can more precisely explore the essential features and patterns hidden in seismic data via intelligent data analysis and pattern recognition. Currently, machine learning applications in seismology focus mainly on two aspects. Firstly, they are used for preliminary earthquake magnitude, timing, and location predictions. Wang et al. [16] constructed a two-dimensional input long short-term memory (LSTM) network capable of learning the correlations between earthquakes at different locations and times and used it for predictions. Their results indicated that this system could make accurate predictions at different temporal and spatial scales. Using multiple seismic activity parameters, Bikash Sadhukhan et al. [17] utilized deep-learning techniques to establish a correlation model between calculated seismic indicators and potential seismic events. This model may predict earthquake magnitudes at different locations and has shown significant and positive results for earthquakes ranging from 3.5 to 6.0 magnitude. Secondly, machine learning can be applied to address complex problems, such as analyzing and interpreting precursory information related to earthquakes. Erman et al. [18] proposed a multi-network-based hybrid long short-term memory (N-LSTM) for ionospheric anomaly detection. This model had good prediction accuracy and stability and successfully detected two total electron content (TEC) anomalies before the Nepal earthquake. Xiong et al. [19] utilized machine learning to identify electromagnetic precursory disturbances in DEMETER data, comparing machine learning algorithms and selecting LightGBM for optimal performance in identifying electromagnetic precursory disturbances before earthquakes. The results indicated that the electromagnetic precursory data within the seismogeneic zone, calculated using the Dobrovolsky formula and a time window of approximately a few hours before the earthquake, provided an effective discrimination of electromagnetic precursory disturbances. Draz et al. [20] investigated multi-parameter precursors with different physical properties, including sea surface temperature (SST), air temperature (AT), relative humidity (RH), outgoing long-wave radiation (OLR), and TEC using standard deviation (STDEV), wavelet transformation, and LSTM networks to identify potential pre- and post-earthquake anomalies. Each method identified noticeable abnormal changes in atmospheric and ionospheric precursors before and after earthquakes. Their research demonstrates the significant relevance of machine learning techniques in detecting seismic anomalies, supporting further studies on the lithosphere–atmosphere–ionosphere coupling (LAIC) mechanism.

Based on previous experience and existing foundations, we chose five commonly used time series forecasting models in this study: one traditional parametric model, autoregressive integrated moving average; two machine learning models, support vector machine (SVM) and extreme gradient-boosting (XGBoost); and two deep learning models, the LSTM and the bi-directional long short-term memory (BILSTM) model. The prediction performance of the models was evaluated with the help of the root mean square error (RMSE) as a metric of accuracy. The results show that the LSTM model exhibited the best performance. Therefore, in this paper, the LSTM model was chosen to predict and analyze the OLR time series anomalies in the 90 days before the Madoi $M_s$ 7.4 earthquake as an example of studying the precursor information of an earthquake. The experimental results were satisfactory, indicating that the pre-earthquake OLR time series analysis based on the LSTM prediction model had good predictive capability and broad application prospects.

The rest of this paper is organized as follows: Section 2 outlines the tectonic context of the seismogenic region and the HIRS/4 OLR-18 data product used in this paper. Section 3 describes the principles of the different time series prediction models and the anomaly

discrimination and extraction methods. Section 4 tests the prediction performance of the various models, based on which the anomaly discrimination and extraction results are obtained. These results are discussed in Section 5, and Section 6 presents the conclusions and outlook for future work in this area.

## 2. Data and Data Preprocessing

### 2.1. Study Earthquakes

The Madoi earthquake occurred in the Bayanhar block in the north-central part of the Qinghai–Tibet Plateau, one of the most representative active blocks of lateral extrusion. The seismogenic fault is the NW-striking Jiangcuo fault, dominated by sinistral strike–slip movement.

The "SERIES OF EARTHQUAKE CASES IN CHINA" is important scientific material for studying Chinese earthquakes and exploring earthquake predictions [21–25]. In the compilation of the series, the statistics of fixed-point anomaly data are distributed as follows: $M_S \geq 7.0$ earthquake, within 500 km; $6 \leq M_S < 7.0$ earthquake, within 300 km; and $5.0 \leq M_S < 6$ earthquakes, within 200 km. Therefore, to fully consider the impact range of the Madoi earthquake, 25 pixels centered on the epicenter at $5° \times 5°$ (32°–37°N, 96°–101°E) were selected as the focus area (Figure 1) to study the preseismic OLR anomalies of the Madoi earthquake.

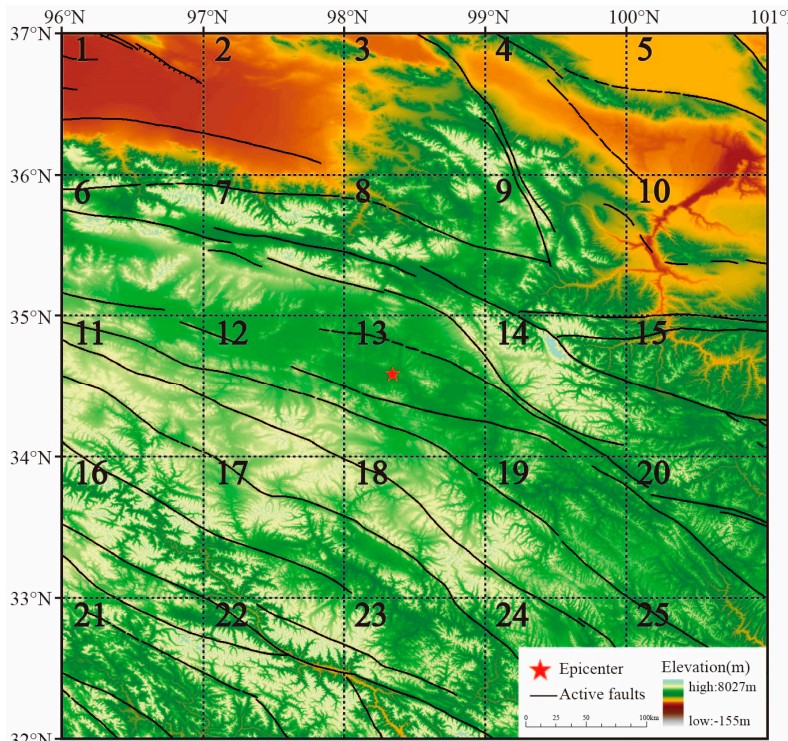

**Figure 1.** Sketch map of geological structure in the study area. The study area consists of 25 grids, denoted by the numbers 1 to 25.

### 2.2. Data

The OLR data were calculated by matching the infrared channel (10.5–12.5 μm) data from NOAA satellites with the total measurements in the broadband (4–50 μm) acquired via the Large Meteorological Experiment Satellite (NIMBUS). The OLR is closely related to factors, such as the temperature and humidity in the atmosphere, and mainly reflects radiation information from cloud tops. Since its launch in 1974, several decades of OLR data have been accumulated, which are of high completeness and continuity and can be downloaded free of charge via NCEP's FTP server (ftp://ftp.cpc.ncep.noaa.gov/precip/noaa18_1x1/) (accessed on 20 March 2023). In this paper, the night-time OLR data from

the NOAA-18 satellite were selected to minimize the influence of solar radiation and human activities on the experimental results, with a temporal resolution of 1 day, a spatial resolution of $1° \times 1°$, and a data unit of $W/m^2$. Each data file comprises $360 \times 180$ global grid points (Figure 2).

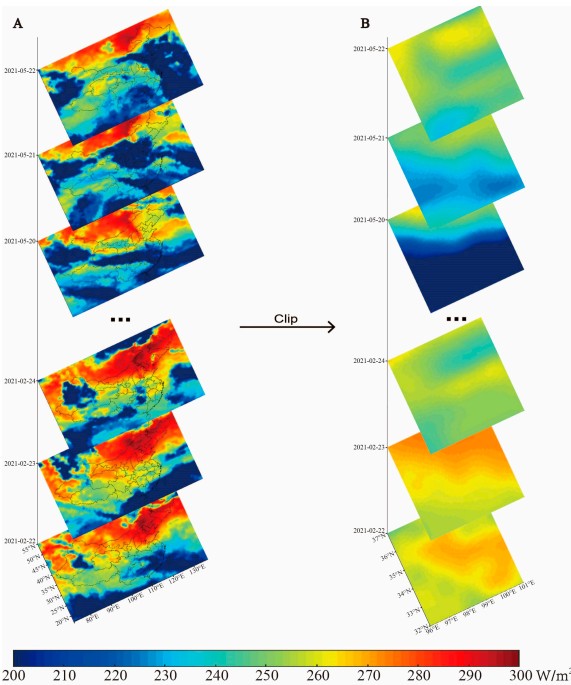

**Figure 2.** (**A**) OLR values at the China scale 90 days before the Madoi earthquake (22 February 2021–22 May 2021). (**B**) OLR values within the $5° \times 5°$ spatial extent of the epicenter of the Madoi earthquake cropped out from A.

### 2.3. Data Preprocessing

Limited by the physical cycle of solar–terrestrial activity, the OLR emitted by the Earth shows periodic changes, especially seasonal changes [26]. To thoroughly study these characteristic variations and minimize the influence of factors such as climate variations, weather patterns, human activities, and vegetation coverage, a five-year OLR time series from 22 May 2016 to 22 May 2021 was selected. At the same time, we noticed these conclusions: Lu et al. [27] studied the variation characteristics of the thermal infrared anomalies of 20 moderately strong earthquakes in Tibet and found that the longest time of long-wave radiation anomalies before the earthquake was 90 days, and the post-earthquake anomalies corresponded well with earthquakes of magnitude five or above; Song et al. [28] found that the thermal anomaly began to gather in space about three months before the Wenchuan earthquake; and Eleftheriou et al. [29] studied dozens of earthquakes with magnitudes ranging from 4.0 to 7.9, and they believed that earthquake anomalies could only be considered to be related to earthquakes if they occurred within 30 days after the earthquake. Therefore, to accurately train the time series prediction model when there is no earthquake, the data of 90 days before and 30 days after the two earthquakes of magnitude five or above that occurred during this period (the $M_s$ 5.3 earthquake in Qinghai Province on 6 May 2018, and the $M_s$ 5.6 earthquake in Shiqu, Sichuan Province on 1 April 2020), totaling 240 days, were deleted. The remaining data were divided into a training set and a test set. The training set included the data from the start time to 90 days before the earthquake, which was used to learn the OLR time series change law and verify the model's performance without an earthquake. The test set was the data obtained 90 days before the earthquake. The singular values in the data were replaced by the average of the same period over the years.

## 3. Methods

Based on the OLR data, a time series prediction model was selected to predict the OLR values in the pre-earthquake epicenter's 5° × 5° spatial range. After preprocessing the OLR data, the training and selection of the model was the key to obtaining the subsequent analysis data, and it was also the core of the whole experiment. The trained model was used to predict the OLR values for ninety days when there was no earthquake and ninety days before the earthquake, respectively, and the similarity test was performed with the actual values to detect whether there was any anomalous period before the earthquake. Finally, 95% confidence intervals were set to extract and analyze the temporal and spatial characteristics of the anomalies to detect the possible precursor information of earthquakes. Figure 3 shows the whole process of the experiment.

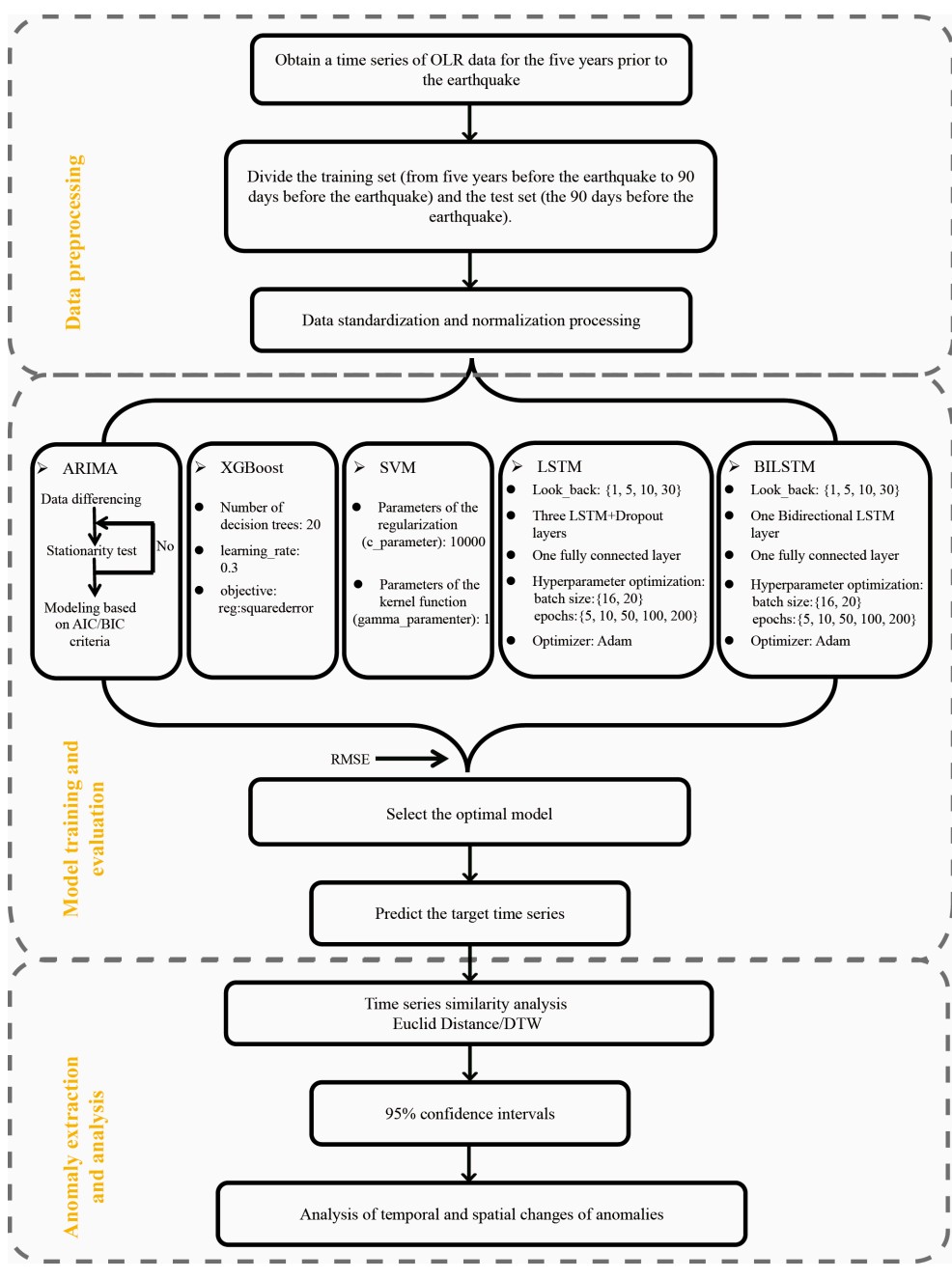

**Figure 3.** Flow chart of the experiment.

### 3.1. Time Series Forecasting Models

3.1.1. ARIMA

The ARIMA model is a famous time series forecasting method proposed by Box and Jenkins in the early 1970s [30]. The model is relatively simple. It treats the data series of the forecasting object over time as a random sequence, describes this sequence approximately by fitting a mathematical model, and predicts the future values based on the time series of past and present values. The ARIMA ($p$, $d$, $q$) model is described as follows:

$$y_t = \varphi_1 y_{t-1} + \cdots + \varphi_p y_{t-p} + \varepsilon_t + \theta_1 \varepsilon_{t-1} + \cdots + \theta_q \varepsilon_{t-q} \tag{1}$$

where $p$ is the order of the autoregressive model, $d$ is the number of differences that make the series smooth, $q$ is the order of the sliding average model, $y_t$ is the value of the OLR time series, $\varphi_i$ is the autoregressive coefficient, $\theta_i$ is the moving average coefficient, and $\varepsilon_t$ is the white noise process.

3.1.2. SVM

SVM is a supervised learning method in machine learning that transforms multiple features nonlinearly via the kernel function, aiming to extract more information from multiple feature inputs and obtain more accurate prediction results [31]. The specific mathematical description is as follows:

The hyperplane is $y = \omega^T x + b$, where $\omega$ and $b$ represent the weight vector and bias, and $T$ denotes the matrix transposition operation. The sample distance of the hyperplane to different classes was maximized according to the SVM principle, which is expressed as

$$arg\ max\left\{ \frac{1}{\parallel \omega \parallel} \cdot min[(\omega^T x + b) \cdot y_i] \right\} \tag{2}$$

where $\parallel \omega \parallel$ represents the L2 norm of the weight vector used to balance the model's complexity and classification accuracy.

According to the distance formula from the point in space to the plane, the distance $d$ from the sample point to the hyperplane can be obtained:

$$d = (\omega^T x + b) \cdot \frac{1}{\parallel \omega \parallel} \tag{3}$$

At this point, the optimization objective of the SVM algorithm is obtained after the introduction of the penalty factor $C$ and the relaxation variable $\xi$:

$$\min_{\omega, b}\ \frac{1}{2} \parallel \omega \parallel^2 + C \sum_{i=1}^{l} \xi_i \tag{4}$$

$$s.t. y_i \left( \omega^T x + b \right) \gg 1 - \xi_i \tag{5}$$

The traditional solution is to introduce Lagrange multipliers:

$$\frac{1}{2} \sum_{i=1}^{l} \sum_{j=1}^{l} \propto_i \propto_j y_i y_j k(x_i x_j) - \sum_{i=1}^{l} \propto_i \tag{6}$$

$$s.t. \sum_{i=1}^{l} \propto_i y_i = 0,\ 0 \leq \propto_i \leq C,\ i = 1, 2, \cdots, l \tag{7}$$

where $k\ ()$ is the kernel function of the SVM algorithm.

### 3.1.3. XGBoost

XGBoost is an optimized distributed gradient boost library that can train the model quickly and more efficiently [32]. The objective function during training consists of two parts: the first part is the gradient boost algorithm loss (Formula (8)), and the second part is the regularization term (Formula (9)).

$$L(\varnothing) = \sum_{i=1}^{n} l\big((y_i', y_i) + f_k(x_i)\big) + \Omega(f_k) \tag{8}$$

where $n$ is the number of training function samples, $l$ is the loss for a single sample, assuming it is a convex function, $y_i'$ is the model's predicted value for the training sample, $y_i$ is the true labeled value of the training sample, and $f_k(x_i)$ represents the predicted value of the decision tree $k$ for sample $i$.

The regularization term defines the complexity of the model:

$$\Omega(f_k) = \gamma T + \frac{1}{2}\lambda \sum_{j=1}^{T} \omega_j^2 \tag{9}$$

where $\gamma$ and $\lambda$ are manually set parameters, $\omega$ is the vector formed by the values of all leaf nodes of the decision tree, and $T$ is the number of leaf nodes.

### 3.1.4. LSTM/BILSTM

LSTM is a popular RNN structure that can learn long-term dependent information [33]. The LSTM network generally consists of three gate units (forgotten gate $f_t$, input gate $i_t$, and output gate $o_t$) and a memory cell (Figure 4). The forgetting gate is responsible for receiving the $h_{t-1}$ output from the hidden layer at the previous time, the $x_t$ newly input, and determining the information to be forgotten, the input gate is responsible for controlling the update and storage of the information, the output gate determines the information to be output in the current state, and the memory cell controls the transmission of the information.

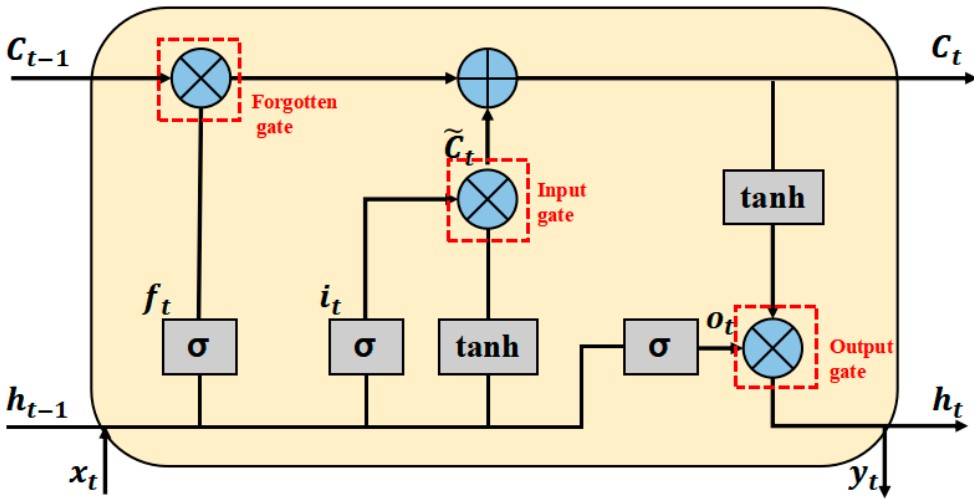

**Figure 4.** LSTM unit structure diagram.

Based on LSTM, BILSTM combines the information of the input sequence in both forward and backward directions. For the output at time *t*, the forward LSTM layer has information for time t and before in the input sequence, and the backward LSTM layer has information for time *t* and after in the input sequence (Figure 5) [34]. The vectors output by the two LSTM layers can be added, averaged, or concatenated.

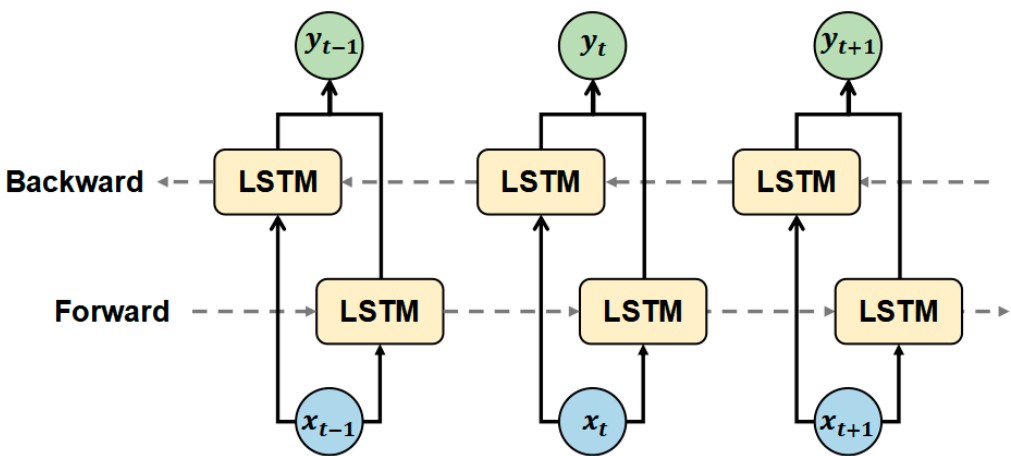

**Figure 5.** BILSTM unit structure diagram.

### 3.2. Model Performance Comparison

This study evaluates the model's predictive performance using the root mean square error (RMSE), which measures the deviation between the observed and predicted values. RMSE, a commonly used error test, has been widely used in artificial intelligence earthquake prediction research [18,35,36]. The formula for calculating RMSE is as follows:

$$RMSE = \sqrt{\frac{1}{N}\sum_{t=1}^{N}(OLR_P - OLR_A)^2} \tag{10}$$

where $N$ is the number of observations involved in evaluating the model, $OLR_P$ is predicted OLR values, and $OLR_A$ is the actual OLR value.

### 3.3. Time Series Similarity Analysis

The historical data of 90 days without the earthquake (1 September 2020–29 November 2020) and 90 days before the earthquake (21 February 2021–22 May 2021) were selected to predict the two OLR time series, respectively, using the model with the highest prediction accuracy after comparison with Section 3.2. The similarity test index uses the Euclidean distance (ED) and dynamic time warping (DTW). The ED and DTW between the predicted and actual values in the two periods were calculated, respectively, to analyze whether there was an abnormal period before the earthquake.

#### 3.3.1. ED

ED is the most common distance measure, measuring the absolute distance between two points in a multi-dimensional space. When the ED is used to compare two time series, a one-to-one correspondence is established between each point in the sequence, and the distance between every two points is calculated and then summed. The smaller the distance, the higher the similarity. The formula for calculating ED is as follows:

$$ED = \sqrt{\sum_{i=1}^{n}(x_{i1} - x_{i2})^2} \tag{11}$$

where $x_{i1}$ represents the $i$-dimensional coordinate of the first point and $x_{i2}$ represents the $i$-dimensional coordinate of the second point.

### 3.3.2. DTW

DTW is a new similarity measurement method that can find the best matching path between data in two arbitrary long time series by adjusting the corresponding relationship between time points. It is robust to noise and can effectively measure the similarity of time series. Suppose first there are two time series, X and Y, of length m and n, respectively. Define a distance matrix D of size $m \times n$, where $D(i, j)$ denotes the distance between the *i*th element of X and the *j*th sum element of Y. Then, define a cumulative matrix C, also of size $m \times n$, where $C(i, j)$ represents the minimum cumulative distances from the first element of X to the ith element and from the first element of Y to the *j*th element. The formula for DTW is derived as follows:

$$C(1,1) = D(1,1) \tag{12}$$

$$C(i,1) = D(i,1) + C(i-1,1), \ 2 \le i \le m \tag{13}$$

$$C(1,j) = D(1,j) + C(1,j-1), \ 2 \le j \le n \tag{14}$$

$$C(i,j) = D(i,j) + min(C(i-1,j), C(i,j-1), C(i-1,j-1)), \ 2 \le i \le m, \ 2 \le j \le n \tag{15}$$

Ultimately, the DTW distance or similarity can be derived from $C(m, n)$. The smaller values of $C(m, n)$ indicate higher similarity between two time series, while larger values indicate lower similarity.

### 3.4. Abnormal Identification and Extraction

A confidence interval refers to the estimated interval of the population parameter constructed by the sample statistics, which shows the degree of confidence that the parameter's actual value falls around the measured value. Besides being often used in statistical research, it has also been effectively applied in discriminating pre-earthquake anomalies [37–39]. Based on these results, we calculated residuals from the raw data and obtained 95% confidence intervals for the predicted values. The difference between the predicted value and the actual value of the time series prediction model is calculated, and the difference between the predicted and actual values is counted as an anomaly when the difference exceeds 95% of the confidence interval. When the difference is higher (or lower) than the upper (or lower) limit of the confidence interval, the actual value is lower (or higher) than the predicted value, and it is regarded as a cold anomaly (or a hot anomaly). The temporal and spatial variation characteristics of the anomalies are then analyzed.

## 4. Results

### 4.1. Model Performance

After determining the optimal parameters for each method, we compared the performance of the different methods. Table 1 presents the prediction performance of each model for 25 grid OLR time series. As shown in Table 1, there were significant differences between the different prediction models. The two deep-learning algorithms performed the best in this experiment, where the LSTM algorithm had the smallest sum of RMSE and the best prediction performance, and the BILSTM algorithm performed second only to the LSTM algorithm. The two machine learning algorithms (SVM and XGBoost) performed equally well next to the deep-learning algorithm, while the traditional parametric model ARIMA had the worst prediction performance. In critical situations, such as detecting OLR data anomalies, relying on models with high uncertainty and low stability does not make sense, so LSTM was chosen as the research algorithm for this experiment.

**Table 1.** Error results based on time series prediction model at different grid points.

| Grid | ARIMA | SVM | XGBoost | LSTM | BILSTM |
|------|-------|-----|---------|------|--------|
| 1 | 34.2619 | 35.1532 | 30.8762 | 30.5294 | 33.6217 |
| 2 | 53.1949 | 36.1017 | 34.9021 | 30.7129 | 32.3657 |
| 3 | 58.2728 | 33.0449 | 31.8505 | 26.7471 | 27.9391 |
| 4 | 44.444 | 30.2318 | 28.8857 | 26.7325 | 27.6086 |
| 5 | 44.444 | 30.2318 | 28.8857 | 26.7325 | 27.6086 |
| 6 | 50.9688 | 40.6984 | 35.6925 | 33.1588 | 34.6644 |
| 7 | 53.0276 | 34.4316 | 36.2105 | 32.0643 | 33.0467 |
| 8 | 47.2451 | 34.6952 | 35.5667 | 30.8911 | 31.451 |
| 9 | 53.2237 | 32.278 | 33.349 | 30.1235 | 29.7703 |
| 10 | 54.5762 | 35.9806 | 38.6714 | 32.6847 | 32.0692 |
| 11 | 53.5756 | 40.6919 | 39.8142 | 36.5470 | 37.1474 |
| 12 | 53.5756 | 40.6919 | 39.8142 | 36.5470 | 37.1474 |
| 13 | 70.4604 | 38.8847 | 36.4824 | 33.1946 | 34.5191 |
| 14 | 41.412 | 40.2266 | 37.6286 | 36.5276 | 36.032 |
| 15 | 41.7221 | 41.2979 | 41.9942 | 36.7073 | 36.7855 |
| 16 | 44.5678 | 40.9163 | 37.4803 | 35.4564 | 34.5752 |
| 17 | 58.5795 | 42.6388 | 37.4096 | 35.6641 | 37.3662 |
| 18 | 52.4613 | 39.7556 | 37.5302 | 34.8795 | 37.4875 |
| 19 | 47.4852 | 37.6338 | 37.0222 | 36.4709 | 37.1824 |
| 20 | 39.4942 | 39.1195 | 40.4514 | 37.6639 | 36.3644 |
| 21 | 61.4502 | 41.8658 | 40.5014 | 38.3213 | 37.3602 |
| 22 | 61.4502 | 41.8658 | 40.5014 | 38.3213 | 37.3602 |
| 23 | 52.6675 | 35.1916 | 34.3085 | 31.4399 | 33.413 |
| 24 | 42.6541 | 32.3148 | 32.4968 | 31.4682 | 32.2569 |
| 25 | 38.0416 | 32.2304 | 34.0134 | 31.7403 | 31.5389 |
| Total | 1253.2563 | 928.1726 | 902.3391 | 831.3261 | 846.6816 |

*4.2. Similarity Comparison*

4.2.1. ED Comparison

The results of the ED values between the actual and predicted time series of OLR for the 25 grids around the epicenter for the 90 days when there was no earthquake and the 90 days before the earthquake are shown in Table 2. As shown in Table 2, the ED values for the 25 grids when there was no earthquake were all lower than those of the pre-earthquake ED values. The two sets of results were processed as differences to visualize the similarity between the two sets of time series and explore the possible regional range of anomalies. The results show that the difference in ED was more significant in the east (grids 14 and 15) and south (grids 17–25) of the epicenter and smaller in the west (grids 11 and 12) and north (grids 1 and 3–12) of the epicenter. The results of the ED comparison reflect abnormal changes in the actual time series before the earthquake. At the same time, the more significant differences in the eastern and southern parts of the epicenter indicate that anomalies were more likely to occur in this region.

**Table 2.** ED between actual and predicted time series.

| Gird | ED in the Aseismic Period | ED in the Seismic Period | Difference Value |
|------|---------------------------|--------------------------|------------------|
| 1 | 186.8928 | 215.2030 | 28.3102 |
| 2 | 182.2871 | 243.9496 | 61.6625 |
| 3 | 198.2106 | 222.8911 | 24.6805 |
| 4 | 193.0780 | 222.0978 | 29.0198 |
| 5 | 193.0780 | 222.0978 | 29.0198 |
| 6 | 200.2466 | 218.4609 | 18.2143 |

**Table 2.** *Cont*.

| Gird | ED in the Aseismic Period | ED in the Seismic Period | Difference Value |
|---|---|---|---|
| 7 | 193.9372 | 231.1959 | 37.2587 |
| 8 | 200.1804 | 239.3742 | 39.1938 |
| 9 | 200.1804 | 239.3742 | 39.1938 |
| 10 | 210.2695 | 231.7173 | 21.4478 |
| 11 | 205.3519 | 218.5727 | 13.2208 |
| 12 | 216.1626 | 243.7730 | 27.6104 |
| 13 | 210.8625 | 254.2664 | 43.4039 |
| 14 | 193.5884 | 244.6240 | 51.0356 |
| 15 | 193.5884 | 244.6240 | 51.0356 |
| 16 | 193.7733 | 216.9230 | 23.1497 |
| 17 | 199.9631 | 259.6670 | 59.7039 |
| 18 | 188.1746 | 253.0382 | 64.8636 |
| 19 | 188.1746 | 253.0382 | 64.8636 |
| 20 | 214.1401 | 287.3047 | 73.1646 |
| 21 | 222.7757 | 282.6755 | 59.8998 |
| 22 | 217.8504 | 294.7557 | 76.9053 |
| 23 | 180.2219 | 262.8554 | 82.6335 |
| 24 | 180.2219 | 262.8554 | 82.6335 |
| 25 | 210.8567 | 296.3231 | 85.4664 |

4.2.2. DTW Comparison

Table 3 shows the DTW values of the actual and predicted time series of the OLR for the 25 grids around the epicenter for 90 days with no earthquake and 90 days before an earthquake. Table 3 shows that the DTW values in the 90 days at the time of no earthquake were all smaller than the DTW of 90 days before the earthquake. Again, the difference between the two was processed, and the results showed that the difference results were more prominent for the grids south of the epicenter. The DTW comparison results show that the similarity between the actual time sequence and the predicted time sequence of the aseismic period was better than that before the earthquake. This further indicates that pre-earthquake signals may occur within the 90 days before an earthquake.

**Table 3.** DTW between actual and predicted time series.

| Gird | DTW in the Aseismic Period | DTW in the Seismic Period | Difference Value |
|---|---|---|---|
| 1 | 1048.3101 | 1122.6331 | 74.323 |
| 2 | 1192.8760 | 1458.1691 | 265.2931 |
| 3 | 1233.5800 | 1460.4359 | 226.8559 |
| 4 | 1201.0691 | 1287.7728 | 86.7037 |
| 5 | 1201.0691 | 1287.7728 | 86.7037 |
| 6 | 1252.1896 | 1292.0052 | 39.8156 |
| 7 | 1218.4646 | 1584.6326 | 366.168 |
| 8 | 1159.1498 | 1524.3408 | 365.191 |
| 9 | 1159.1498 | 1524.3408 | 365.191 |
| 10 | 1328.1454 | 1423.4520 | 95.3066 |
| 11 | 1306.5641 | 1342.0472 | 35.4831 |
| 12 | 1258.7495 | 1451.5585 | 192.809 |
| 13 | 1262.1207 | 1613.9854 | 351.8647 |
| 14 | 1150.4503 | 1752.4018 | 601.9515 |
| 15 | 1150.4503 | 1752.4018 | 601.9515 |
| 16 | 1213.2118 | 1244.7485 | 31.5367 |
| 17 | 1120.0947 | 1756.5435 | 636.4488 |

**Table 3.** *Cont.*

| Gird | DTW in the Aseismic Period | DTW in the Seismic Period | Difference Value |
|---|---|---|---|
| 18 | 983.54445 | 1500.6782 | 517.13375 |
| 19 | 983.54445 | 1500.6782 | 517.13375 |
| 20 | 1331.7225 | 1870.4373 | 538.7148 |
| 21 | 1252.8426 | 2052.9131 | 800.0705 |
| 22 | 1285.1753 | 2056.8329 | 771.6576 |
| 23 | 990.76198 | 1797.7720 | 807.01002 |
| 24 | 990.76198 | 1797.7720 | 807.01002 |
| 25 | 1326.1863 | 1954.9361 | 628.7498 |

*4.3. Abnormal Temporal and Spatial Characteristics*

4.3.1. Temporal Scale Characteristics

Figure 6 shows the anomalies in the temporal dimension. According to our definition of hot and cold anomalies in Section 3.4, it is clear that all the pre-seismic anomalies of this predicted Madoi earthquake were "hot" anomalies. The anomalies appeared in 13 periods during the 90 days before the earthquake. The earliest anomaly appeared on 13 March, with a small amplitude. The first concentration of anomalies occurred on 18 March, covering six grids, and another anomaly of smaller magnitude occurred on 31 March. There were three anomalies on 14 April, 18 April, and 24 April, with three grids simultaneously having anomalies on 18 April. After entering May, the frequency of anomalies increased, and the anomaly amplitude increased. On May 6, three grids were abnormal, and the maximum radiation energy exceeded the lower limit of 17.05. On 9 May, the anomaly range was the largest, 14 grids were abnormal at the same time, and the maximum radiation energy exceeded the lower limit of 24.17. Four days later, the concentrated anomaly occurred again, 11 grids were abnormal, the intensity of the anomaly increased further, and the maximum radiation energy of six grids exceeded the lower limit of 30. On the next day, the concentrated anomaly still existed, but the scope and intensity decreased; seven grids were abnormal, and the maximum radiation energy was 21.62. There were three more minor anomalies on 16 May, 17 May, and 20 May, respectively. In general, the closer to the time of earthquake occurrence, the greater the anomaly intensity, the wider the coverage area, and the anomaly development trend is appearing–strengthening–weakening–disappearing.

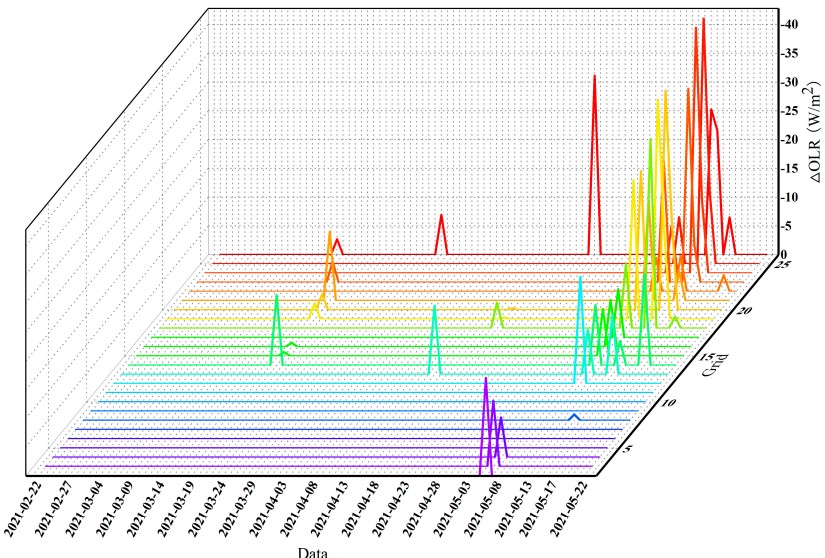

**Figure 6.** OLR anomaly prediction before the Madoi earthquake—temporal scale. The anomalies in different grids are represented by different colors.

### 4.3.2. Spatial Scale Characteristics

Figure 7 shows the anomalies of 25 grids in the spatial dimension, 7 of which did not exceed the confidence interval, and these grids were mainly distributed in the northern and northeastern regions of the epicenter. Before the earthquake, some slight anomalies occurred northwest of the epicenter. Anomalies with strong intensity and large areas were mainly concentrated in the south and southeast of the epicenter. Combined with Figure 1, we found more fault zones in the south of the epicenter, while fewer fault zones were in the north. Therefore, within one month before the Madoi earthquake, the anomalies were mainly distributed in the south and southeast of the epicenter with more fault zones, and we speculate that the location of frequent anomalies may have had a particular relationship with the activity of the fault zone.

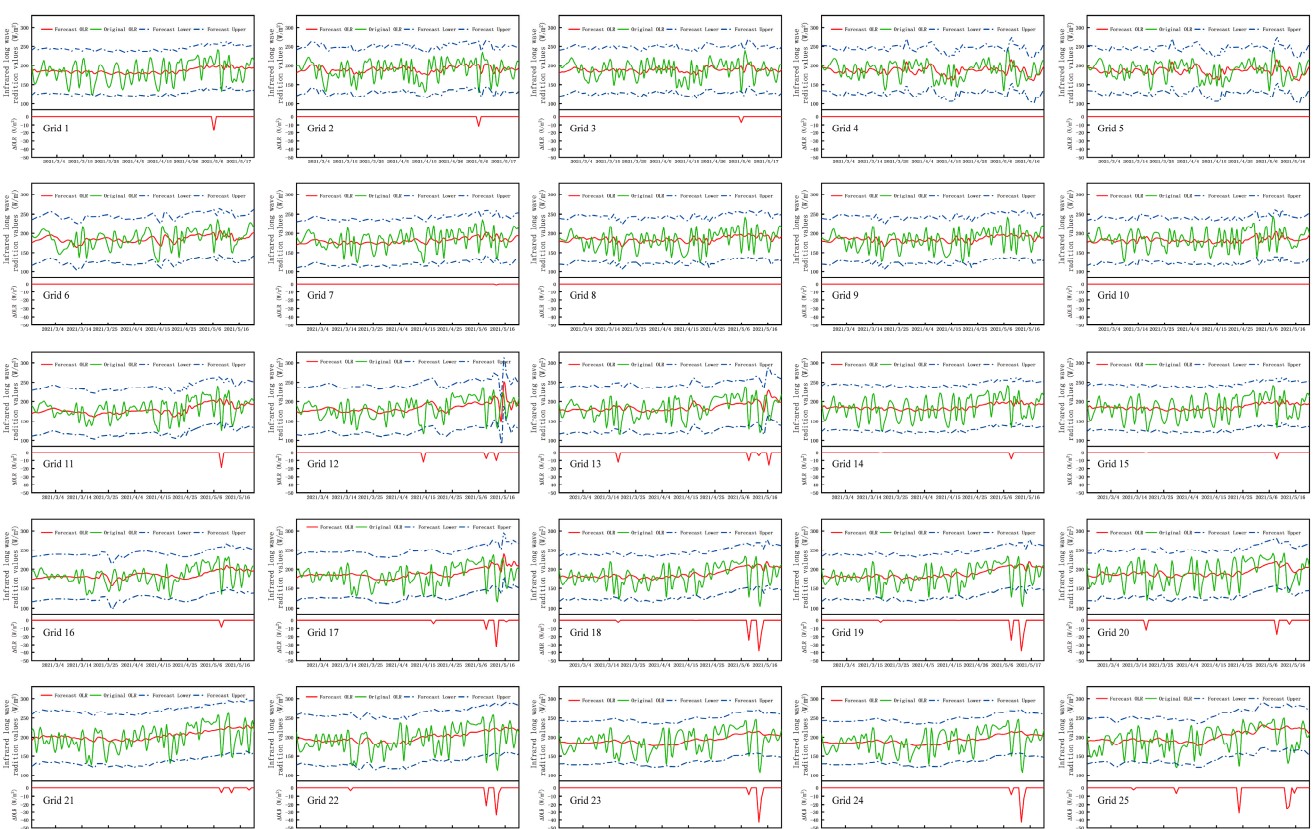

**Figure 7.** OLR anomaly prediction before the Madoi earthquake—spatial scale.

## 5. Discussion

### 5.1. Consistency of LSTM Results with Traditional Methods

Tramutoli introduced the Absolutely Local Index of Change of the Environment (ALICE) method [40], which relies on the robust satellite technology (RST) approach for detecting pre-seismic thermal infrared anomalies. This method constructs a background field based on multi-year observational data to depict the state of thermal infrared anomalies during periods of seismic quiescence. It has demonstrated valuable applications in several earthquake events [41,42]. To validate the effectiveness of the LSTM model in detecting preseismic OLR anomalies, we conducted a comparative analysis between the predicted anomalies and those extracted using the conventional ALICE method. The ALICE formula is defined as

$$ALICE(x, y, t) = \frac{V(x, y, t) - \mu(x, y)}{\sigma(x, y)} \tag{16}$$

In the equation, $ALICE(x, y, t)$ represents the anomaly value at time $t$ for the coordinate position (x, y) of a pixel; $V(x, y, t)$ represents the pixel value at time $t$ for the coordinate

position (x, y); $\mu(x, y)$ represents the multi-year average at the same location and time; and $\sigma(x, y)$ represents the corresponding standard deviation.

The ALICE method was employed to select a research area encompassing a $10° \times 10°$ region with the epicenter of the Madoi earthquake at its center. The chosen time frame spanned the 90 days preceding the earthquake, aligning with the time window used in the LSTM method for predicting preseismic anomalies. This approach was adopted to acquire insights into the evolution of OLR anomalies leading up to the Madoi earthquake, subsequently generating spatial distribution maps of OLR anomalies (Figure 8). According to a previous study, an ALICE value greater than 2 was considered to fall within the category of anomalies [43].

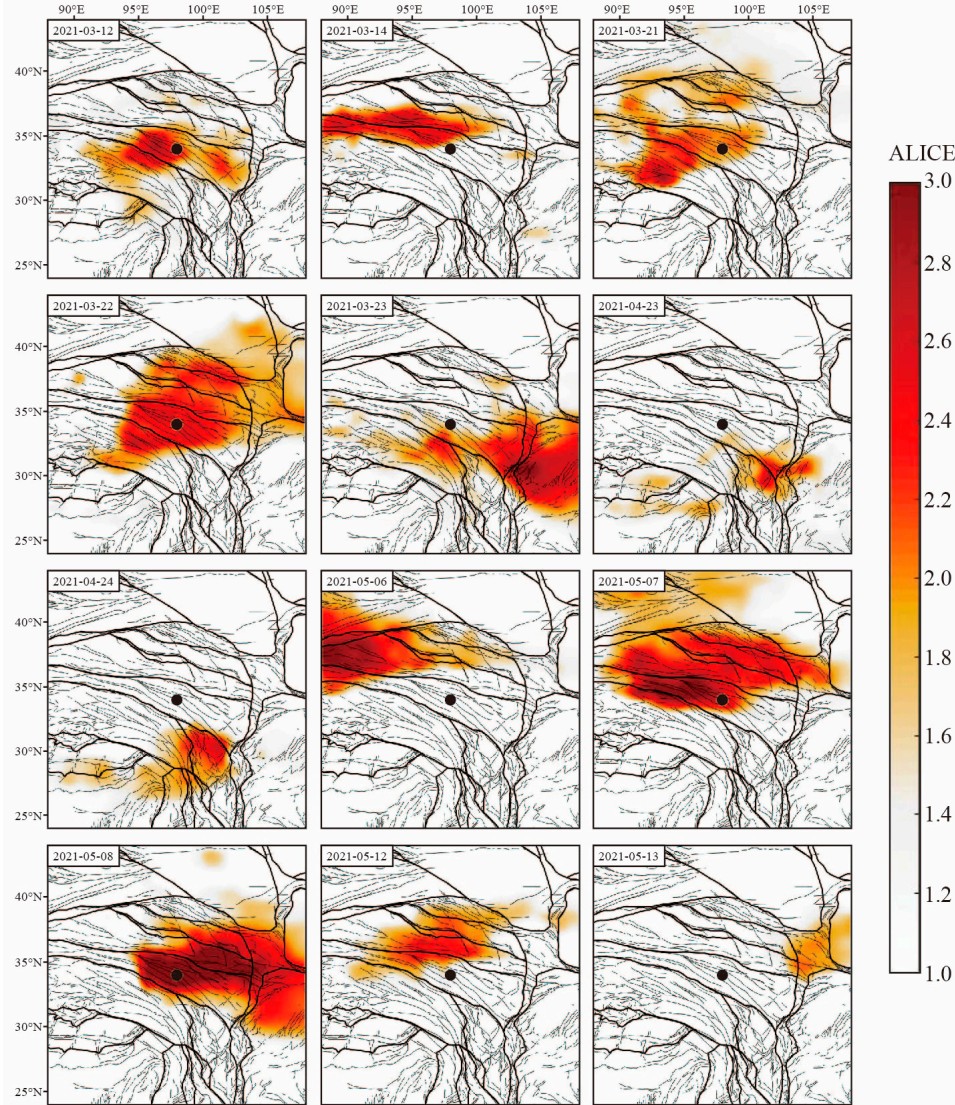

**Figure 8.** Daily scale thermal infrared anomaly of the Madoi earthquake. The black dots represent the location of the epicenter of the Madoi earthquake, the thick black lines represent different blocks, and the thin black lines represent active faults.

From Figure 8, it is evident that the ALICE method successfully detected significant preseismic OLR anomalies. In total, there were 12 instances of anomalies observed. The first anomaly occurred on 12 March near the epicenter. The anomaly reappeared on March 14th in the form of strips covering the seismic fault zone. Starting from 21 March, the anomaly shifted eastward, expanding its coverage during this process and lasting for three days. Subsequently, a period of calm persisted for a month. Within the last 30 days

preceding the earthquake, the anomalies' frequency, extent, and intensity increased. Seven anomalies occurred within this time frame. On 23 April, an anomaly moved from the southeast to the south of the epicenter and lasted for two days. On 6 May, an anomaly appeared northwest of the epicenter and then moved eastward along the seismogenic fault zone, approaching the epicenter. On 8 May, the anomaly peaked, covering the epicenter with an ALICE value of 3 before disappearing. On 12 May, an anomaly emerged in the northwest of the epicenter, lasting for two days, and disappeared from the eastern part of the epicenter. Comparing the temporal characteristics of anomalies extracted using the ALICE method with the predictions from the LSTM method shown in Figure 6, we observe a strong correspondence regarding the occurrence time, frequency, and anomaly intensity. This suggests that utilizing the LSTM method to detect preseismic OLR anomalies is trustworthy to a certain extent and holds research value for practical applications.

*5.2. Robustness of the LSTM Model*

To verify whether the model still maintains good prediction performance for earthquake precursor signals under different spatial and temporal conditions, we tested the model using information from six earthquakes and two hypothetical random earthquakes (A and B) that occurred in mainland China in recent years. Table 4 summarizes and organizes this information. Figure 9 illustrates the preseismic OLR anomaly extraction results for these earthquake examples.

**Table 4.** Earthquake information used in our study.

| Date (y-m-d) | Longitude (°E) | Latitude (°N) | Depth (km) | Magnitude ($M_S$) | Location |
|---|---|---|---|---|---|
| 2022-01-08 | 101.26 | 37.77 | 10 | 6.9 | Menyuan, Qinghai |
| 2021-05-21 | 99.87 | 25.67 | 8 | 6.4 | Yangbi, Yunnan |
| 2021-03-19 | 92.74 | 31.94 | 10 | 6.1 | Biru, Tibet |
| 2020-06-26 | 82.33 | 35.73 | 10 | 6.4 | Yutian, Xinjiang |
| 2019-04-24 | 94.61 | 28.40 | 10 | 6.3 | Medog, Tibet |
| 2017-08-09 | 82.89 | 44.27 | 11 | 6.6 | Jinghe, Xinjiang |
| 2019-02-28 | 112 | 44 | — | — | A |
| 2019-01-31 | 83 | 40 | — | — | B |

The following information can be obtained from Figure 9: Before the Menyuan earthquake, six anomalies were observed. The first anomaly occurred on the 79th day before the earthquake; then, there were two centralized anomalies occurring on the 80th day and the 54th day before the earthquake, and the last anomaly centrally appeared on the 32nd day before the earthquake and then tended to be calm until the earthquake occurred. There were eight days of anomalies before the Yangbi earthquake, with the earliest anomaly occurring on the 60th day before the earthquake and the last anomaly occurring on the 9th day before the earthquake, and almost every anomaly appeared on multiple grids at the same time. There were also eight days of anomalies before the Biru earthquake, with the earliest anomaly occurring on the 85th day before the earthquake, the last anomaly occurring on the 5th day before the earthquake, and anomalies occurring on 13 grids at the same time. For the Yutian earthquake, anomalies first appeared 87 days before and continued until the 2nd day before, totaling 22 days with anomalies. On the 50th day before, 18 grid cells exhibited anomalies simultaneously; at other times, anomalies were mostly concentrated in multiple grid cells. There were 14 days of anomalies before the Medog earthquake, with the first anomaly occurring on the 84th day before the earthquake and the last anomaly occurring on the 2nd day before the earthquake. The fluctuation in anomalies before the earthquake was obvious, with concentrated anomalies occurring for many days. The earliest anomaly of the Jinghe earthquake occurred on the 82nd day before the earthquake, and several strong and concentrated anomalies appeared from the 80th to the 70th day before the earthquake, with the earthquake occurring on the 4th day after the last concentrated anomaly ended.

Overall, the model successfully detected significant OLR anomaly changes within 90 days before all six actual earthquakes. The anomalies exhibited several distinctive features: firstly, the anomalies had a widespread impact, often simultaneously affecting multiple grid cells. Secondly, the anomalies displayed substantial intensity, frequently exceeding the confidence interval by more than 30. Thirdly, the anomalies were recurrent, with multiple anomalies observed within the 90 days before the earthquakes. On the other hand, the anomalies extracted for hypothetical earthquakes A and B were very quiet, with occasional minor anomalies, but these fluctuations did not correspond to the three characteristic phenomena mentioned above. Therefore, we believe that this model demonstrates robustness in exploring seismic precursors. It performed well in extracting and tracking pre-earthquake abnormal signals.

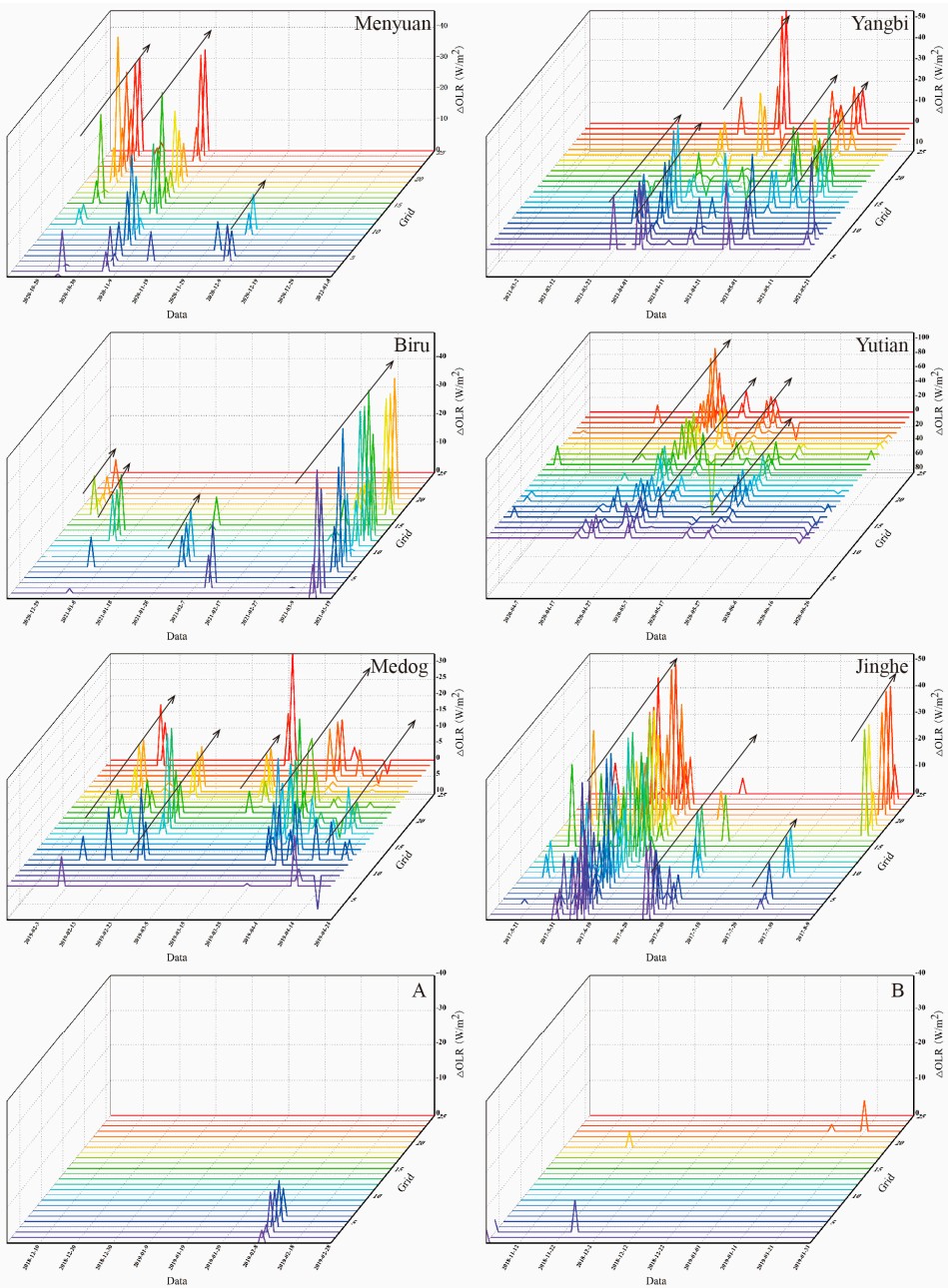

**Figure 9.** Extraction results of preseismic OLR anomalies for actual earthquakes and hypothetical random earthquakes. The black arrow indicates a concentrated anomaly. The anomalies in different grids are represented by different colors.

### *5.3. Advantages of the LSTM Model*

Finding anomalies in any time series data is a very critical task. Based on statistical theory, some anomaly discrimination methods have proven the existence of infrared long-wave anomalies before earthquakes: Kang et al. [44] analyzed the thermal and infrared change characteristics before and after 14 November 2001, west of the Kunlun Mountain Pass magnitude 8.1 earthquake, based on the OLR data of NOAA satellites and the brightness temperature data of the Japanese geostationary meteorological satellite GMS-5. The results showed that the warming anomaly in OLR occurred one month before the earthquake, the brightness temperature anomaly occurred one week before the earthquake, and the anomaly characteristics were consistent with the tectonic distribution of the earthquake region in the spatial pattern. Zhang and Meng [43] conducted long-term statistical studies on the relationship between thermal infrared anomalies and earthquakes in different regions using multi-year nightly OLR data from NOAA satellites and Molchan plots. The results show that TIR anomalies were highly correlated with magnitude 4.0 normal fault or reverse fault earthquakes and that TIR anomalies induced by earthquakes should be spatially and temporally persistent. Although these methods have achieved good results in earthquake monitoring and forecasting, the results may be potentially different due to the setting of anomaly discrimination rules affected by a variety of factors [45,46], and it is difficult to summarize a universal method for studying different seismic cases. The LSTM algorithm belongs to the data-driven method, which fully considers the time correlation of the time-series data itself and utilizes historical data for training and prediction based on the data of the current time step so that LSTM can achieve end-to-end training and prediction on time series data without manually extracting features or performing other preprocessing steps, which can reduce the error caused by human or environmental factors' interference, and the applicability is high. In addition, the traditional OLR anomaly extraction algorithm mainly focuses on hot anomalies [14,47]. Still, this experiment subdivides the types of anomalies by setting confidence intervals, which can capture cold and hot anomalies simultaneously, to facilitate the study of the generation mechanism of anomalies. For example, Han et al. [48] studied the strain evolution characteristics around the epicenter for many years before the Madoi earthquake, and the results showed that, in the early stages of the earthquake, the block where the epicenter was located was in the deformation gradient zone of tension and pressure conversion, which made it easy to cause energy accumulation, and was a seismic risk area. At the same time, considering that, according to rock stress experiment results, a rock is heated via extrusion and cooled via tension [49], we speculate that the anomaly type may be able to predict the regulation of active faults on rock stress, which provides a new method for exploring the relationship between anomaly generation and tectonic activity.

### *5.4. Multi-Parameter Coupling Relationship and Anomaly Generation Mechanism Analysis before Earthquake*

According to the LAIC model, complex variations occur before an earthquake. A review of the research literature related to the Madoi earthquake reveals several significant findings: Du and Zhang [50] analyzed ionospheric disturbances in the seismogenic area before and after the Madoi earthquake. They noticed that anomalous ionospheric disturbances began to appear about 40 days before the earthquake. About 20 days before the earthquake, the electron density of Ne changed dramatically, and these anomalies disappeared after the earthquake. Zhang et al. [51] discovered multiple occurrences of electromagnetic anomalies within the last two weeks before the earthquake. These anomalies had short intervals between adjacent events and a wide-ranging impact. They suggested a potential connection between these anomalies and the occurrence of the Madoi earthquake. Wang et al. [52], using the Benioff strain as a response parameter, studied the evolution of load–unload response ratio (LURR) anomalies within 400 km of the Madoi earthquake epicenter. They observed that the LURR values gradually increased during the two months leading up to the earthquake, reaching their highest point approximately one month before the event. Afterward, they gradually decreased, indicating that the rock medium in the

seismogenic area had reached the end of its yielding stage. Su et al. [53] observed that the water level near the epicenter's observation station dropped significantly on February 25 and the $HCO_3^-$ concentration in the observation well water had continued to rise since March. Considering that it was caused by increased carbon dioxide dissolution in the water, they thought that it might indicate an increase in fault activity in this area. Jing et al. [54] analyzed the spatial and temporal evolution of the preseismic thermal variations in the Madoi earthquake using the Index of Microwave Radiation Anomaly (IMRA) based on eight years of microwave brightness temperature (MWBT) data. Changes in MWBT are directly dependent on the surface and subsurface temperature and emissivity. Their results showed that enhanced IMRA had been distributed along the seismogenic fault since mid-February, with longer-lasting and stronger anomalies reoccurring in March and April 2021. They also analyzed near-surface CO variations detected via AIRS sensors, noting two significant CO anomalies on 23 February and 23 March. They suggested that these might be indicative of accelerated gas release due to changes in tectonic stress. Luo et al. [55] studied the water radon values observed at the Huangyuan station near the Madoi earthquake epicenter. Their findings revealed a distinctive pattern of the water radon values from April to June, characterized by a "decline (during the seismic event)–sustained low values–rebound–recovery" anomaly. This pattern was similar to the short-term anomalies observed in water radon values before and after several recent earthquakes. They also noted a sharp drop in water radon measurements three days before the earthquake, suggesting that water radon value changes could serve as short-term precursory anomalies for the Madoi earthquake.

The chronological sequence of these anomalies is illustrated in Figure 10. Building upon the findings from these multi-parameter studies and following the LAIC concept [56], an attempt was made to construct a coupled parameter model for the Madoi earthquake, as depicted in Figure 11. The accumulation of stresses in the crustal medium of the seismogenic zone before the earthquake could have some effects: (1) Some of the rocks composing the crust carry electrical charges. These charges exist as electron vacancies in the valence band and typically remain stable. Under the influence of stress, positive holes (p-holes) in the valence band are activated, impacting the rock's conductivity and altering its geophysical and chemical properties, generating electromagnetic radiation, thermal infrared radiation, and more. (2) Stress accumulation induces thermoelastic effects in rocks, causing them to heat up due to friction. (3) Stress accumulation leads to the fracturing of crustal rocks, creating a connected network for subsurface gases' (such as radon) transportation, resulting in a continuous seepage of gases from the rocks. (4) Stress accumulation also induces changes in the preseismic geophysical environment. These alterations significantly impact the atmospheric electric field, ultimately leading to abnormal variations in electron density (e.g., Ne) within the ionosphere. Furthermore, both positive charge vacancies and escaping radon can trigger air ionization, which has an important effect on long-wave radiation. Therefore, our interpretation of the preseismic OLR anomalies related to the Madoi earthquake suggests that they may result from multiple mechanisms acting in concert, including processes like greenhouse gas degassing and electromagnetic field anomalies. Moreover, because changes in long-wave radiation anomalies represent the integral of all thermal effects related to the earthquake, from the Earth's surface to the top of the troposphere, encompassing variations in surface temperature, air temperature, and latent heat flux, OLR exhibits more extensive anomaly changes compared with other parameters.

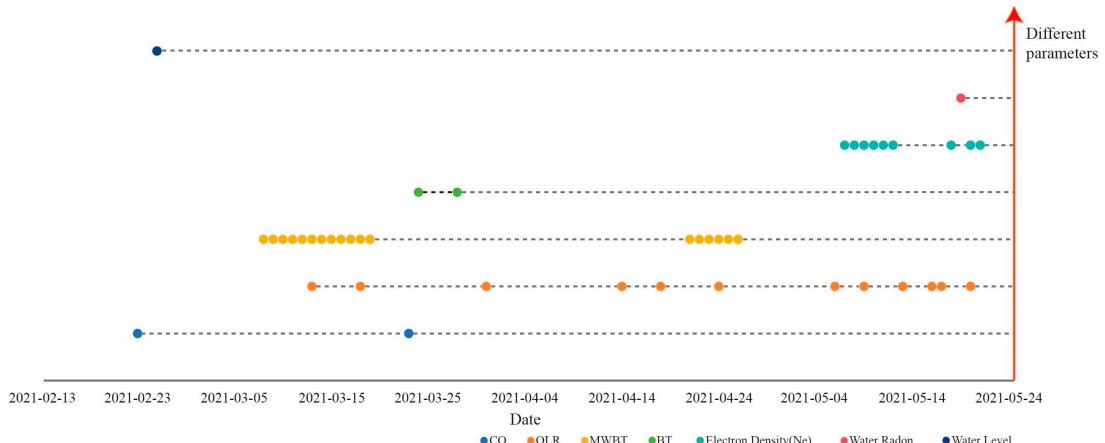

**Figure 10.** Time distribution of multi-parameter anomalies in Madoi earthquake.

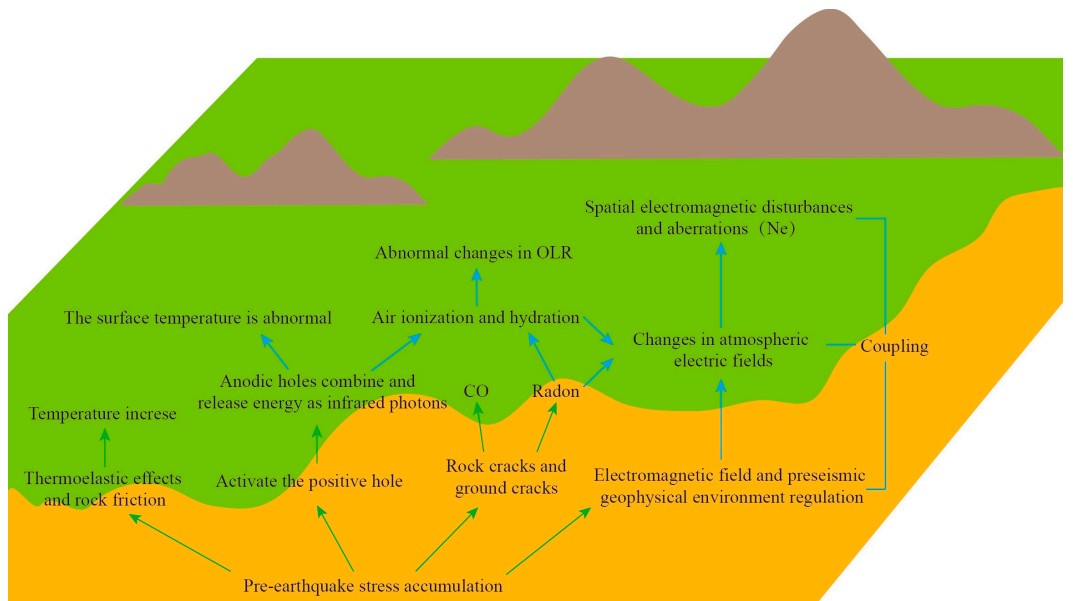

**Figure 11.** Diagram of Madoi earthquake multi-parameter coupling model. The green arrows represent underground processes and the blue arrows represent surface and atmospheric processes.

## 6. Conclusions

Based on the OLR data, the precursory anomalies of the Madoi earthquake are preliminarily explored in this paper. The LSTM model with the best performance is selected to learn the variation law of the OLR time series without an earthquake, and the OLR values in two periods (90 days in the aseismic period and 90 days before the earthquake) before the Madoi earthquake are predicted, respectively. Based on the evaluation of ED and DTW time series similarity methods, the similarity degree between the predicted value and the actual value in the two periods is quantified, which is used to determine whether there is an abnormal period within the 90 days before the earthquake. Finally, the 95% confidence interval is used to extract and track the spatiotemporal features of the anomalies. The following conclusions are drawn:

(1) The two time series similarity analysis methods proved that the similarity between the time series of the actual values at the time of the aseismic and the predicted values were better than those before the earthquake, indicating an anomalous period.

(2) The results of the RMSE-based accuracy evaluation show that, among the five selected time series prediction models, the prediction performance of the LSTM model is better than several other time series models, which is attributed to the strong learning ability

of the deep-learning model on data features. After accepting the input time series data, the LSTM layer, utilizing its ability to process the contextual information and the memory capability, has the potential to dig in and capture the potential nonlinear features in the time-series data.

(3) The anomalies are extracted, and the temporal and spatial change characteristics of the anomalies are analyzed using 95% confidence intervals. On the time scale, the earliest anomalies appeared seventy days before the earthquake. The closer to the moment of the earthquake, the higher the frequency of anomalies, the wider the coverage area, and the higher the intensity; the anomalies exceeded the lower limit of the confidence interval of 40 W/m$^2$ on the ninth day before the earthquake. On the spatial scale, the anomalies are concentrated in the southern and southeastern parts of the epicenter, where the distribution of fault zones is denser, and are less frequent in the northern part of the epicenter, where the fault zones are sparse, which suggests that tectonic activity is potentially related to the occurrence of earthquakes and the emergence of radiation anomalies. In addition, it is worth noting that the selection of the confidence interval affects the extraction of anomalies, and as the threshold setting of the confidence interval increases, some small amplitude chance anomalies unrelated to earthquakes will be filtered out, and the results will be more reliable.

(4) The method of exploring the pre-earthquake signals of the Madoi earthquake with deep-learning prediction OLR time series is feasible. The application of the method achieved the expected purpose. After the occurrence of anomalies, we should focus on the rupture zones and faults in a locking state around the relevant geological bodies, which may be advantageous locations for the origin of future strong earthquakes. At present, only a complete experiment has been carried out for the Madoi earthquake, and this method will be applied to the processing and analysis of other moderate and strong earthquakes in the future to further verify the feasibility of the method of exploring pre-earthquake signals by predicting OLR time series with deep learning.

**Author Contributions:** Conceptualization, J.Z. (Jingye Zhang) and K.S.; methodology, J.Z. (Jingye Zhang); software, J.Z. (Jingye Zhang) and J.Z. (Junqing Zhu); validation, J.Z. (Jingye Zhang) and K.S.; formal analysis, J.Z. (Jingye Zhang); investigation, N.M.; resources, J.Z. (Jingye Zhang); writing—original draft preparation, J.Z. (Jingye Zhang); writing—review and editing, J.Z. (Jingye Zhang), D.O. (Dimitar Ouzounov) and N.M.; visualization, J.Z. (Junqing Zhu); supervision, K.S.; project administration, K.S.; funding acquisition, K.S. All authors have read and agreed to the published version of the manuscript.

**Funding:** The work of J.Z., K.S., J.Z. and N.M. was funded by the National Natural Science Foundation of China under Grant No. U2039202, the National Key Research and Development Program of China under Grant No. 2019YFC1509202, and the Central Public-Interest Scientific Institution Basal Research Fund No. CEAIEF2022050500. The work of D.O. for this manuscript has been supported by the Institute of ECHO at Chapman University research funds.

**Institutional Review Board Statement:** Not applicable.

**Informed Consent Statement:** Not applicable.

**Data Availability Statement:** This study uses NOAA-18 satellite $1° \times 1°$ OLR data, which can be downloaded via NCEP's FTP server (ftp://ftp.cpc.ncep.noaa.gov/precip/noaa18_1x1/) (accessed on 20 March 2023).

**Acknowledgments:** We thank the National Natural Science Foundation of China and the National Key Research and Development Program of China for funding this study. We thank NOAA for providing the OLR data and Sha Yin for their guidance in producing the figures.

**Conflicts of Interest:** The authors declare no conflict of interest.

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
