# Peer review of "Application of Model-Based Time Series Prediction of Infrared Long-Wave Radiation Data for Exploring the Precursory Patterns Associated with the 2021 Madoi Earthquake"

_remotesensing, doi:10.3390/rs15194748_

Round 1

Reviewer 1 Report

This paper have compared Five-time series prediction models, including autoregressive integrated moving average (ARIMA), support vector machine (SVM), extreme gradient boosting (XGBoost), long short-term memory (LSTM) and bi-directional long short-term memory (BILSTM). LSTM with the highest prediction accuracy was selected to retrospectively predict the OLR values during Madoi MS 7.4 earthquake and a period before the earthquake in the area. The results showed that the two-time series similarity analysis methods proved that the similarity between the time series of the actual values at the time of the aseismic and the predicted values were better than before the earthquake. Then the temporal and spatial distribution characteristics of the anomalies in the 90 days before the earthquake were analyzed with the 95% confidence interval as the criterion of the anomalies: anomalies of the different grids appeared on similar dates, and the anomalies of high values appeared centrally in the time of the earthquake. I recommend this article for publication on Remote Sensing with the following minor revisions:

1. Figures 4 and 5 need to be optimized, and font types and sizes in Figure 4 need to be aligned. The resolution of Figure 5 is so poor that only the waveform can be seen, but the vertical and horizontal axes cannot be recognized.

2. Figure 8 is not mentioned or cited in the text. And Figure 8 still needs to be improved. Only the near-surface temperature anomalies and the long-wave radiation anomalies are represented in the figure, but what about the other anomalies? For example, ionospheric electron density anomalies, CO anomalies, water radon anomalies, and so on. The connection between various parameters in Figure 8 also needs to be described and discussed in detail in the paper.

Author Response

Dear Esteemed Reviewer,

Greetings! We have incorporated the modifications based on your valuable feedback and have submitted them in the form of an attachment for your review. The details of the revisions can be found in the attached document.

Best regards.

Reviewer 2 Report

The authors proposed to use the time series prediction models to predict the outgoing long-wave radiation (OLR) anomalies and study short-term earthquake precursors. Five-time series prediction models, including autoregressive integrated moving average (ARIMA) and long short-term memory (LSTM), were trained with the OLR time series data. The calculation results indicated that the precursors of the earthquake have been detected. But the paper needs to be revised in the following aspects:

1. The application status of machine learning in earthquake precursor information needs to be added in the introduction.

2. The training set should consider the impact of spatial weather changes on data.

3. The process of model training and how to set parameters should be described in detail.

4. Should the difference between the predicted value and the actual value satisfy a normal distribution.

5. Random earthquakes should be added to test the stability of the model.

Suggestion: Review after modification.

The English writing proficiency of this manuscript still needs to be improved, especially in terms of sentence structure, terminology, and other aspects. There is still room for improvement.

Author Response

(The authors gave the same response as above.)

Reviewer 3 Report

       The presented manuscript is devoted to the prediction of the outgoing long-wave radiation (OLR) anomalies with time series models to study short-term earthquake precursors. Taking May 21, 2021, Madoi MS 7.4 earthquake as an example, five-time series prediction models, including autoregressive integrated moving average (ARIMA) and long short-term memory (LSTM), were trained with the OLR time series data of the aseismic moments in the 5° × 5° spatial range around the epicenter. It was found by comparing the predicted value time series with the actual value time series that the similarity indexes of the two-time series before the earthquake were lower than that of the aseismic period, indicating that the predicted time series before the earthquake significantly differed from the actual time series. All the models are described in sufficient details. The obtained results are of undoubted interest. Nevertheless, I have indicated some major and minor remarks (see enclosed file), which do not allow me to give immediately a positive assessment of this work.

          To summarize, the new approach proposed by authors is promising and may be applied. I think that this manuscript could be interesting for a potential reader of the "Remote Sensing" journal. Nevertheless, the presented work has not devoid of some shortcomings. That is why, I believe that this work can be reconsidered again after major revision.

Minor editing of English language required.

Author Response

Dear Esteemed Reviewer,

Greetings! We have incorporated the modifications based on your valuable feedback and have submitted them in the form of an attachment for your review. The details of the revisions can be found within the attached document.

Best regards.

Round 2

Reviewer 3 Report

I think, that the revised manuscript can be accepted for publication in the present form.

Author Response

Dear Esteemed Reviewer,

Greetings! Thank you for recognizing our research work. Your invaluable suggestions have significantly improved the quality of our manuscript. Once again, we appreciate the time and effort you dedicated to reviewing our paper.

Best regards,

Jingye Zhang